# Efficient photocatalytic production of hydrogen peroxide using dispersible and photoactive porous polymers

Shengdong Wang[1,2], Zhipeng Xie[3], Da Zhu[4], Shuai Fu [5], Yishi Wu[6], Hongling Yu[3], Chuangye Lu[2], Panke Zhou[3], Mischa Bonn [5], Hai I. Wang [5,7], Qing Liao [6], Hong Xu [4], Xiong Chen [3] ✉ & Cheng Gu [1] ✉

Developing efficient artificial photocatalysts for the biomimetic photocatalytic production of molecular materials, including medicines and clean energy carriers, remains a fundamentally and technologically essential challenge. Hydrogen peroxide is widely used in chemical synthesis, medical disinfection, and clean energy. However, the current industrial production, predominantly by anthraquinone oxidation, suffers from hefty energy penalties and toxic byproducts. Herein, we report the efficient photocatalytic production of hydrogen peroxide by protonation-induced dispersible porous polymers with good charge-carrier transport properties. Significant photocatalytic hydrogen peroxide generation occurs under ambient conditions at an unprecedented rate of 23.7 mmol g$^{-1}$ h$^{-1}$ and an apparent quantum efficiency of 11.3% at 450 nm. Combined simulations and spectroscopies indicate that sub-picosecond ultrafast electron "localization" from both free carriers and exciton states at the catalytic reaction centers underlie the remarkable photocatalytic performance of the dispersible porous polymers.

The efficient production of $H_2O_2$ is vital for applications in science and technology[1-4]. Chemists have attempted to synthesize $H_2O_2$ using sustainable strategies based on photochemistry, such as using inorganic titanium dioxide[5] or polymeric carbon nitride[6] to generate $H_2O_2$ by photocatalytic processes. However, the efficient photocatalytic production of $H_2O_2$ has been challenging because of the poor visible-light absorption and sluggish charge/exciton transport[1]. Donor-acceptor-based linear[7,8] or crosslinked[9-13] polymers have recently been developed and proposed for $H_2O_2$ production because of their structural tunability. Especially, two-dimensional covalent organic

frameworks (COFs)[14-21] and covalent triazine frameworks (CTFs)[22-24] are emerging candidates for efficient photocatalysis because of their predesigned molecular structures and tunable exciton dynamics. However, the localized Frenkel-type excitons in such materials with high exciton binding energy ($E_b$) of hundreds of meV to 1 eV, limits the dissociation and long-range migration of photogenerated charge carriers, posing a critical obstacle in the generation, separation, and transport of charges. A second critical issue in crosslinked porous photocatalysts is that these materials are usually synthesized as insoluble and unprocessable powders[25]. The poor dispersibility of such

[1]College of Polymer Science and Engineering, State Key Laboratory of Polymer Materials Engineering, Sichuan University, 610065 Chengdu, People's Republic of China. [2]State Key Laboratory of Luminescent Materials and Devices, Institute of Polymer Optoelectronic Materials and Devices, South China University of Technology, 510640 Guangzhou, People's Republic of China. [3]State Key Laboratory of Photocatalysis on Energy and Environment, College of Chemistry, Fuzhou University, 350116 Fuzhou, People's Republic of China. [4]Institute of Nuclear and New Energy Technology, Tsinghua University, 100084 Beijing, People's Republic of China. [5]Max Planck Institute for Polymer Research, Ackermannweg 10, 55122 Mainz, Germany. [6]Beijing Key Laboratory for Optical Materials and Photonic Devices, Department of Chemistry, Capital Normal University, 100048 Beijing, People's Republic of China. [7]Nanophotonics, Debye Institute for Nanomaterials Science, Utrecht University, Princetonplein 1, 3584 CC Utrecht, The Netherlands. ✉e-mail: chenxiong987@fzu.edu.cn; gucheng@scu.edu.cn

materials results in interfacial mismatch and inhomogeneous hydrophilicity that restrict the migration/percolation of charge carriers and ionic species between water and photocatalysts[26]. Because of the abovementioned limitations, the $H_2O_2$ generation rate in the state-of-the-art photocatalysts is typically below 2 mmol $g^{-1}$ $h^{-1}$ (Supplementary Table S1). Such a low rate is far from scale-up production satisfaction. Here, we propose a design strategy to construct photocatalysts with efficient charge-carrier generation and transport at the water-catalyst interface.

We achieve efficient photocatalytic production of $H_2O_2$ by designing a highly dispersible and photoactive porous photocatalyst. To solve the dispersion and solution-processing issues of crosslinked porous polymers, we employ ionization on the skeletons of the porous polymers to make them highly dispersed in organic solvents for high-quality thin film preparation[27,28]. We call this the "charge-induced dispersion (CID)" mechanism[29]. To further advance this mechanism, we encode a photoactive moiety and a highly reversible proton acceptor into the skeleton of a covalent triazine polymer (CTP) to manipulate its photoactivity and dispersibility, which we term the "in-situ protonation" strategy. Specifically, we produce a dynamically protonated CTP featuring low exciton binding energy and high charge-carrier mobility. This design accelerates the photocatalytic production of $H_2O_2$ at ambient conditions to a production rate of 23.7 mmol $g^{-1}$ $h^{-1}$ and an apparent quantum efficiency of 11.3% at 450 nm, outperforming most reported $H_2O_2$ photocatalysts.

## Results

### Synthesis and solution processing of the CTPs

To construct the CTPs with photoactivity and dynamic protonation functionality, we synthesized a monomer comprising benzonitrile and thiazolo[5,4-*d*]thiazole (TT) moieties (TT–BN; Supplementary Materials and Figs. S1–S7). The synthesis of the CTP was achieved by a triflic acid (TfOH)-catalyzed trimerization, accompanied by the in-situ protonation of the triazine moiety (Fig. 1a)[30], thereby introducing substantial amounts of positive charges to the skeleton and making the product well dispersed in triflic acid to form a transparent, homogeneous, solution-like sol (Supplementary Fig. S8). Subsequently, the protonated CTP (termed TTH–CTP) was subjected to NaOH solution for the deprotonation of the framework and afforded the neutral product (termed TT–CTP). Both TT–CTP and TTH–CTP could reversibly transform into their counterparts by adding acid and base, respectively (Supplementary Fig. S9). The structures of TT–CTP and TTH–CTP were characterized by various analytic methods to demonstrate the chemical structure, porosity, morphology, amorphous nature, and thermal stability (Supplementary Figs. S10–S16 and Table S2). The protonation degree in TTH–CTF was 93.8% according to the X-ray photoelectron spectroscopy (XPS) analysis (Supplementary Fig. S17 and Table S3).

Remarkably, various highly polar organic solvents, such as *N,N*-dimethylacetamide (DMA), *N*-methyl pyrrolidone (NMP), *N,N*-dimethylformamide (DMF), dimethyl sulfoxide (DMSO), 1,3-dimethyl-2-imidazolidinone (DMI), and acetonitrile (ACN), could readily dissolve TTH–CTP by simple manual shaking, forming yellow-colored, transparent, solution-like sols with a distinct Tyndall effect (Fig. 1b). Among these solvents, TTH–CTP was most soluble in acetonitrile with a solubility of 3.5 mg $mL^{-1}$ (Supplementary Table S4), and the resulting solutions were stable without sediment when left under ambient conditions for over one year. These features sharply contrast the suspensions of traditional crosslinked porous materials, which were non-transparent and unstable. TTH–CTP could even be dispersed in water to form a uniform and stable colloidal suspension (Fig. 1b), which was crucial to generate a semi-homogeneous catalytic system and eliminate the interfacial mismatch in traditional porous materials-water heterogeneous systems[26]. The high solubility of TTH–CTP enabled the production of solution-processed high-quality thin films by spin-coating (Supplementary Fig. S18) with controlled film thickness (Supplementary Fig. S19). Subsequently, the TT-CTP films could be readily produced by immersing the TTH–CTP films in ammonia water (Supplementary Fig. S18). The facile preparation of both CTP films allowed us to measure their photophysical properties accurately, thereby disclosing the underlying mechanisms of photocatalytic performance.

### Photocatalytic production of $H_2O_2$

TT–CTP and TTH–CTP exhibited Ultraviolet–Visible (UV–Vis) absorption onsets at 526 and 569 nm, corresponding to the optical bandgaps of 2.34 and 2.24 eV, respectively (Supplementary Fig. S20). The Mott-Schottky profiles revealed the flat-band potentials of TT–CTP and TTH–CTP to be −0.53 and −0.41 V, respectively (Supplementary Fig. S21). The band structures of TT–CTP and TTH–CTP estimated by combining optical bandgaps with Mott-Schottky plots indicated that both CTPs are capable of synthesizing $H_2O_2$ (Supplementary Fig. S22). Both CTPs showed similar photoluminescence properties in terms of emission peaks (548 and 566 nm for TT–CTP and TTH–CTP, Supplementary Fig. S23), photoluminescence quantum yields (4.1% and 3.3% for TT–CTP and TTH–CTP, Supplementary Fig. S24), lifetimes (1.61 and 1.91 ns for TT–CTP and TTH–CTP, Supplementary Fig. S25), and outstanding photostability (Supplementary Fig. S26). Both CTPs exhibited photo-induced charge-separation capability with substantial transient photocurrent responses under visible-light illumination (Supplementary Fig. S27), indicating at least part of the optical excitation leads to free charge-carrier generation in CTPs. Electrochemical impedance measurements revealed a slightly smaller semicircle in TTH–CTP (Supplementary Fig. S28), suggesting that ionic moieties promote exciton dissociation and charge-carrier migration.

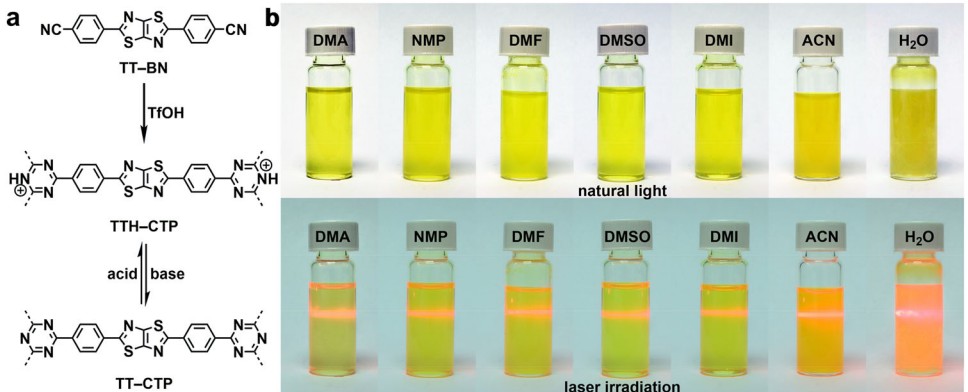

**Fig. 1 | Synthesis and solution processing of the CTPs. a** Synthesis of TT–CTP and TTH–CTP. **b** Photos of TTH–CTP dispersed in various solvents under natural light and laser irradiation, showing the Tyndall effect.

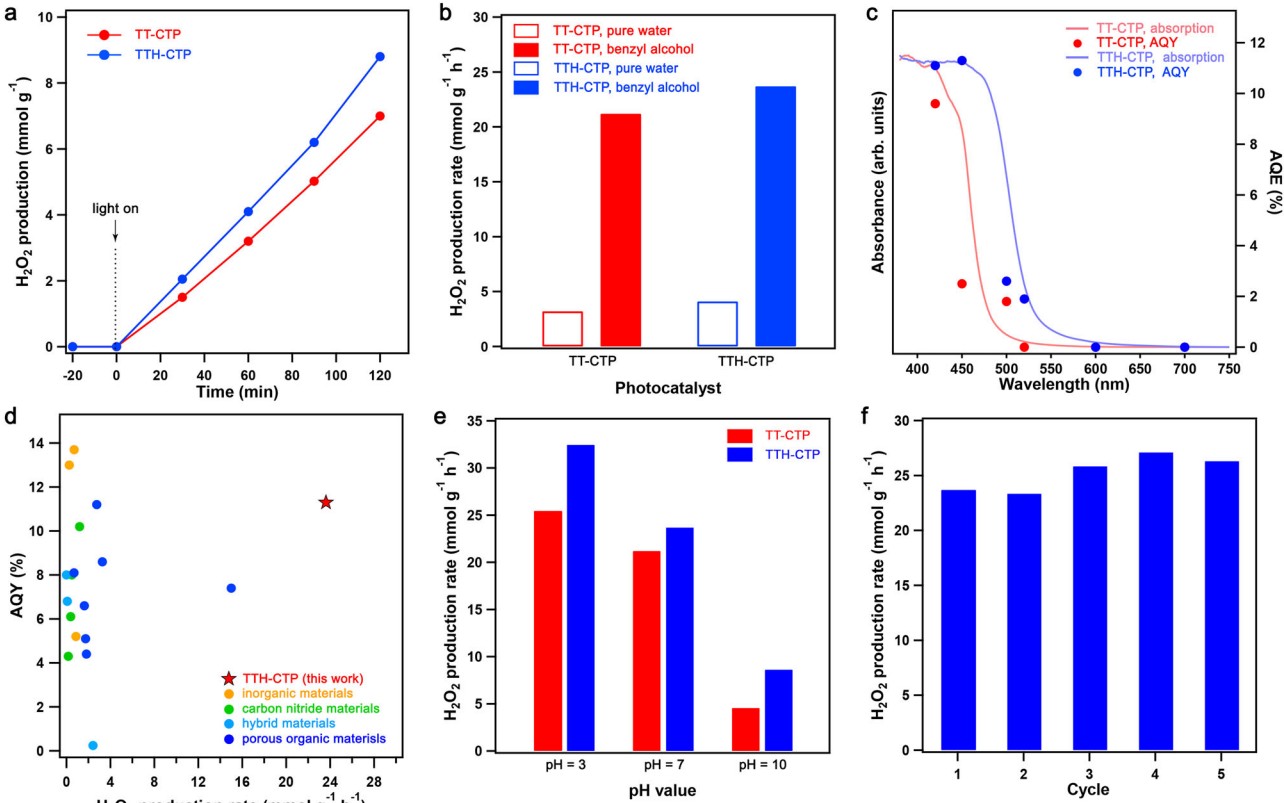

**Fig. 2 | Photocatalytic production of H₂O₂. a** Time-dependent production rate of H₂O₂ on TT–CTP and TTH–CTP in pure water. **b** H₂O₂ generation rates of photo-catalytic half-reaction of TT–CTP and TTH–CTP with and without benzyl alcohol as a hole sacrificial agent. **c** UV–Vis spectra and AQY of H₂O₂ generation for TT–CTP and TTH–CTP. **d** H₂O₂ generation rates and AQY of different photocatalysts. The red star represents TTH–CTP in this work (with benzyl alcohol, see panel **b**) and the orange, green, sky-blue, and blue circles correspond to other previously reported photocatalysts. **e** H₂O₂ generation rates of photocatalytic half-reaction of TT–CTP and TTH–CTP under different pH values. **f** H₂O₂ generation rates of photocatalytic half-reaction of TTH–CTP under different cycles.

The exceptional dispersibility and charge-separation properties of TT–CTP and TTH–CTP inspired us to conduct H₂O₂ photocatalytic production experiments from O₂ and H₂O, initially performed in a batch reactor under Xe-lamp irradiation in pure water (Supplementary Fig. S29). No H₂O₂ was produced using the catalyst-free system or under dark conditions (Supplementary Figs. S30 and S31). By contrast, the generation of H₂O₂ under light irradiation over TT–CTP and TTH–CTP was observed with a linearly increasing concentration with irradiation time (Fig. 2a), yielding the H₂O₂ production rates of 3.2 and 4.1 mmol g⁻¹ h⁻¹, respectively (Fig. 2b). The production rates were substantially increased by adding small amounts of benzyl alcohol as a sacrificial agent; a benzyl alcohol/H₂O = 1/9 (v/v) system afforded the H₂O₂ production rates of 21.2 and 23.7 mmol g⁻¹ h⁻¹ for TT–CTP and TTH–CTP, respectively (Fig. 2b; Supplementary Figs. S32 and S33). By irradiating with monochromatic light, the apparent quantum efficiency (AQE) values of TT–CTP and TTH–CTP were measured to be 9.6% (at 420 nm) and 11.3% (at 450 nm), respectively (Fig. 2c). Such high H₂O₂ production performance in terms of rates and AQEs outperforms most H₂O₂-generation photo-catalysts reported to date (Fig. 2d, Supplementary Table S1), though a direct comparison among different photocatalytic systems is not easy because of varied reaction conditions and equipment. Note that TTH–CTP showed an improved performance than TT–CTP, although the exciton and charge-carrier transport of the latter were slightly more favorable. This was presumably due to the superior dis-persibility of TTH–CTP in the benzyl alcohol/H₂O system, which facilitated the interfacial homogeneity and boosted the charge transport on the water/TTH–CTP/benzyl alcohol interfaces (Supple-mentary Fig. S34). TTH–CTP could catalyze the H₂O₂ generation under air conditions (Supplementary Fig. S35), suggesting the potential for scale-up production.

The monomer TT–BN barely showed catalytic activity (Supple-mentary Fig. S36), revealing the contribution of the highly conjugated CTP skeletons to H₂O₂ production. The pH-dependent photocatalysis experiments revealed the activity followed the order of pH = 3 > pH = 7 > pH = 10 for TT–CTP and TTH–CTP, with a substantially enhanced H₂O₂ production rate of 25.4 and 32.5 mmol g⁻¹ h⁻¹ at pH = 3, respec-tively (Fig. 2e). This indicates that protonation-induced ionic species promote the activity. The contribution of the catalyst amount to the H₂O₂ concentration revealed volcano-type curves with a maximum H₂O₂ production rate of 31.5 mmol g⁻¹ h⁻¹ at the catalyst amount of 25 mg and pH = 7 (Supplementary Fig. S37), suggesting the balanced generation and decomposition of H₂O₂ (Supplementary Fig. S38). TTH–CTP exhibited outstanding stability in a five-run cycling test (Fig. 2f).

To further verify the reaction mechanism in pure water, we per-formed the oxygen evolution reaction (OER) using TT–CTP and TTH–CTP as photocatalysts, respectively. TT–CTP and TTH–CTP exhibited the O₂ generation rates of 1.7 and 2.2 mmol g⁻¹ h⁻¹ (Supple-mentary Fig. S39), indicating that the oxidation and reduction half-reactions in the H₂O₂ generation process were the 4-electron water oxidation and 2-electron O₂ reduction, respectively. To identify the active oxygen species in the reaction, we conducted electron para-magnetic resonance (EPR) spectroscopy using 5,5-dimethyl-1-pyrroline N-oxide (DMPO) as the spin-trap agent. The typical six characteristic signals for DMPO·O₂⁻ were observed in TT–CTP and TTH–CTP under light irradiation, with intensity enhanced with irradiation time (Sup-plementary Fig. S40), indicating the generation of ·OOH intermediate

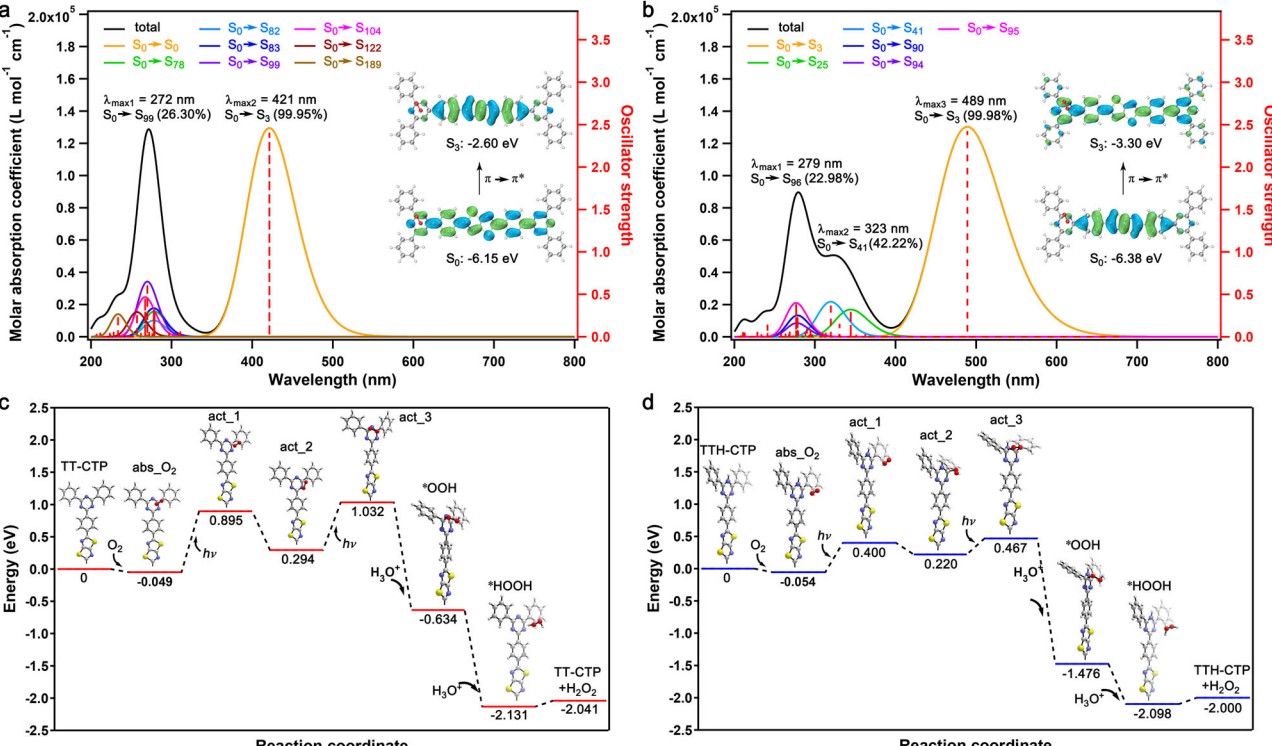

**Fig. 3 | Theoretical calculations. a** TD-DFT-calculated absorption spectra and oscillator strengths for the model system of TT−CTP at the $O_2$-adsorbed state. The insets are the transition orbits at the maximum oscillator strength. **b** TD-DFT-calculated absorption spectra and oscillator strengths for the model system of TTH−CTP at the $O_2$-adsorbed state. The insets are the transition orbits at the maximum oscillator strength. **c** Calculated free energy diagrams of $H_2O_2$ production catalyzed by TT−CTP model system. The state of "abs_$O_2$" denotes the $O_2$-adsorbed state, and the states of "act_1", "act_2", and "act_3" represent three evolutionary steps to activate the $O_2$ molecule. **d** Calculated free energy diagrams of $H_2O_2$ production catalyzed by TTH−CTP model system. The state of "abs_$O_2$" denotes the $O_2$-adsorbed state, and the states of "act_1", "act_2", and "act_3" represent three evolutionary steps to activate the $O_2$ molecule. C: gray; N: blue; H: white; O: red; S: yellow.

species. By adding trapping agents for singlet oxygen, superoxide radical, and free electrons to the reaction system, respectively, the electron transfer and the formation of superoxide radicals were concluded to be the critical step in the $H_2O_2$ generation (Supplementary Fig. S41).

## Theoretical studies

To understand the catalytic mechanism of $H_2O_2$ production, we performed time-dependent density functional theory (TD-DFT) calculations with model systems of TT−CTP and TTH−CTP. To confirm the protonated position on TTH−CTP, we added one proton on TT and triazine moieties, respectively, followed by optimizing their geometrical structures and calculating their total energies. The total energy of the triazine-protonated structure was lower than that of the TT-protonated structure by 1.032 eV (Supplementary Fig. S42), demonstrating that protonation of the triazine moiety was thermodynamically favorable. Electrostatic potential analysis revealed that protonated TTH−CTP is substantially electron-positive compared to neutral TT−CTP (Supplementary Fig. S43), suggesting the stronger adsorption affinity of TTH−CTP to $O_2$. Indeed, the $O_2$-adsorption energy of TTH−CTP was calculated to be larger than that of TT−CTP by 0.808 kJ mol$^{-1}$ (Supplementary Fig. S44).

The highest occupied molecular orbital (HOMO) energy levels of TT−CTP and TTH−CTP were mainly distributed on the TT or TTH moieties, whereas the lowest unoccupied molecular orbital (LUMO) energy levels were delocalized on the whole skeleton (Supplementary Fig. S45), thus forming a combined local and charge-transfer excited state for charge separation upon light irradiation. The calculated bandgaps of TT−CTP and TTH−CTP model systems were 3.55 and 3.08 eV (Supplementary Fig. S46), in good agreement with the UV–Vis

spectra. Accordingly, the simulated energy spectra of TT−CTP and TTH−CTP model systems at initial and $O_2$-adsorbed states were obtained (Fig. 3a, b; Supplementary Figs. S47 and S48). For $O_2$-adsorbed TT−CTP and TTH−CTP, we observed dominant transitions of $S_0 \rightarrow S_3$ with the largest oscillator strengths ($f$) of 2.40 and 2.42, respectively (Fig. 3a, b). Moreover, the electron transition gap was smaller for TTH−CTP than for TT−CTP, indicating that TTH−CTP was more favorable than TT−CTP in photocatalytic $H_2O_2$ production in terms of light absorption. TT−CTP exhibited larger energy barriers in the $O_2$-activation process compared to TTH−CTP (Fig. 3c, d; Supplementary Tables S5 and S26), which was consistent with experimental results that TTH−CTP showed higher photoactivity in the photocatalytic production of $H_2O_2$.

## Discussion

### Charge-separation dynamics

To uncover the exceptional photocatalytic activity from the photochemical aspects, we combined steady and transient spectroscopies to reveal the charge-separation dynamics in the CTPs. We first conducted temperature-dependent photoluminescence studies to estimate the exciton binding energy $E_b$ (i.e., the energy required to ionize excitons into free carriers) of TT−CTP and TTH−CTP films[31]. The photoluminescence intensity increased monotonically with decreasing temperature (Fig. 4a, b, inset) due to the more favorable exciton formation at low temperatures, which in turn also indicated the presence of a large fraction of free carriers at high temperatures. The $E_b$ values of TT−CTP and TTH−CTP films extracted from the fitting of an Arrhenius equation were 74.5 ± 2.0 and 83.2 ± 2.0 meV, respectively (Fig. 4a, b). These values are quite low compared to most organic semiconductors with $E_b$ usually in the range of 500 to 1000 meV (Supplementary

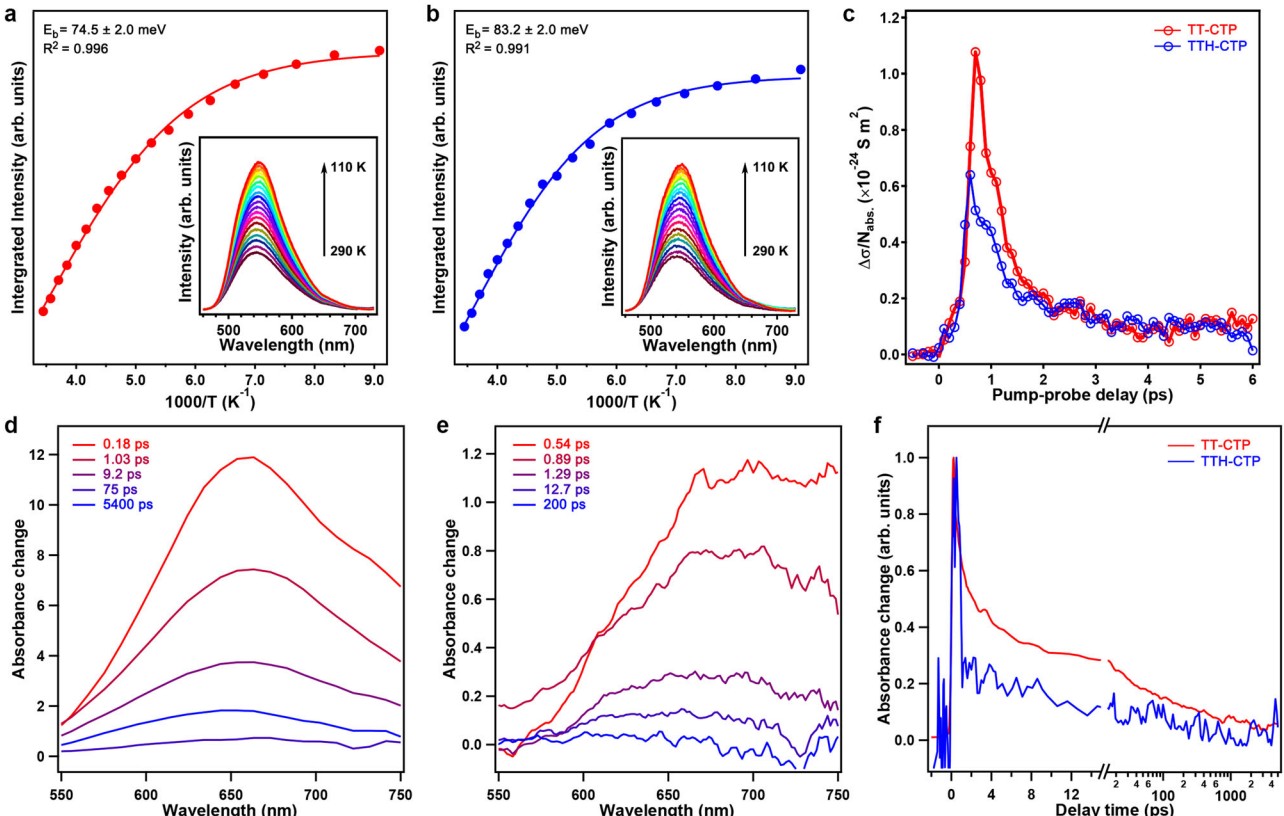

**Fig. 4 | Charge-carrier dynamics of the CTPs. a** Temperature-dependent integrated photoluminescence intensity of TT–CTP. The red line is the fitting curve according to the Arrhenius equation. The inset shows the temperature-dependent photoluminescence spectra. **b** Temperature-dependent integrated photoluminescence intensity of TTH–CTP. The blue line is the fitting curve according to the Arrhenius equation. The inset shows the temperature-dependent photoluminescence spectra. **c** Terahertz photoconductivity dynamics of TT–CTP and TTH–CTP thin films. The samples were photoexcited by a 400-nm femtosecond pump pulse with an incident pump fluence of 886 μJ cm⁻² in a dry N₂ environment. **d** TA spectra for TT–CTP thin film upon excitation at 500 nm. **e** TA spectra for TTH–CTP thin film upon excitation at 500 nm. **f** Comparison of the 660-nm kinetic curve in TT–CTP thin film and the 680-nm kinetic curve in TTH–CTP thin film.

Table S27). The small values of $E_b$ favor the charge separation in TT–CTP and TTH–CTP for photocatalytic reactions.

To further study the charge separation and transport effects in CTPs, we combined time-resolved terahertz spectroscopy (TRTS, see Supplementary Materials)[32] and femtosecond transient absorption spectroscopy (TA, in the visible range) spectroscopy. While TRTS reports free-carrier conductivity and dynamics (in particular for our samples; see discussion later), TA provides exciton population dynamics. For TRTS studies, following the optical generation of charge carriers in CTPs by 3.1 eV excitations, we probed the transport properties of free carriers with a single-cycle ~ps THz field. As shown in Fig. 4c, we attributed the sub-ps rise to the free-carrier generation and conduction in the CTP films, further confirmed in the frequency-resolved complex photoconductivity (e.g., the real part dominated photoconductivity, Supplementary Fig. S49). The free carriers were found to be short-lived, living only 1–2 ps, which could be attributed to the efficient "trapping" of free electrons to the catalytic center. This result is consistent with the "delocalized" nature of electrons and localized holes in the calculated band structure. We further estimated the product of charge-carrier mobility ($\mu$) and free-carrier quantum yield ($\varphi$, $0 \le \varphi \le 100\%$; see Supplementary Materials and Fig. S49), from which we inferred the lower limit of carrier mobility (assuming $\varphi = 100$), $\mu = (6.7 \pm 1.7) \times 10^{-2}$ and $(4.0 \pm 1.1) \times 10^{-2}$ cm² V⁻¹ s⁻¹ for TT–CTP and TTH–CTP films, respectively, typical for organic semiconductors. Furthermore, the photoresponse of the film was approximately 10 times that of its powder form (Supplementary Fig. S50), highlighting the importance of the solution-processed high-quality CTP films for charge-carrier migration.

The TA technique was then carried out on both TT–CTP and TTH–CTP films to investigate the exciton dynamics[33]. Immediately after excitation, the TA spectra of TT–CTP film showed one dominant excited state absorption (ESA) band peaked at 663 nm (Fig. 4d), which corresponds to the absorption of the $S_1 \rightarrow S_n$ transition. A similar but red-shifted spectral pattern (peaked at 680 nm) could be found for TTH–CTP film (Fig. 4e). This redshift behavior of the ESA feature was coincident with that of steady-state absorption data (Fig. 2c). Figure 4f compared the exciton dynamics of TT–CTP and TTH–CTP films. Both the 663-nm and 680-nm kinetic curves can be well described by multi-exponential decays (Supplementary Table S28). In particular, we observed a fast, sub-ps decay in the exciton population (0.4 ps for TTH–CTP vs. 0.6 ps for TT–CTP) for both samples; the spectra weight of this fast decay accounted for 87% of the dynamics for TTH–CTP, which was substantially higher than that of TT–CTP (57%). Here we assigned the fast component to electron trapping to the catalytic center, leading to exciton dissociation. As the fast ~ps decay occurs in both the TRTS and TA results, we excluded the exciton-to-free-carrier or free-carrier-to-exciton conversion as the main mechanism of the decay. Instead, our results indicated that the "localization" of photogenerated electrons (in both free and bound forms) to the catalytic centers not only provides a consistent picture to account for the dynamics observed by TRTS and TA, but also unveils the microscopic transport nature underlying the efficient photocatalytic reaction. The scenario could also explain the more efficient photocatalytic performance of TTH–CTP than that of TT–CTP, thanks to the enhanced electron capture radius (mainly from exciton states) in TTH–CTP.

## Outlook

This study demonstrated efficient photocatalytic production of $H_2O_2$ by designing highly dispersible CTPs with exceptional charge-carrier transport properties. The high dispersibility of the protonated CTP allowed the facile preparation of thin films for disclosing the nature of exciton and charge-carrier migration, yet the compatibility of the water system with the dispersible CTPs facilitated the carrier transport at the water-photocatalyst interface. The low exciton binding energy of the CTPs ensured efficient exciton dissociation, together with decent charge-carrier mobility, making CTPs an outstanding photocatalytic candidate for $H_2O_2$ production with remarkable generation rates and quantum yields. Our rationale could be broadly applied to design porous polymer-based photocatalysts to promote catalytic performance by integrating solution processability with photoactivity.

## Methods

### Synthesis of 4,4'-(thiazolo[5,4-d]thiazole-2,5-diyl)dibenzonitrile (TT–BN)

Dithiooxamide (720 mg, 6.0 mmol) and 4-formylbenzonitrile (1810 mg, 13.8 mmol) were mixed in DMF (20 mL) in a 100-mL Shrek tube under a nitrogen atmosphere in a glove box. The tube was sealed with a rubber stopper and heated at 150 °C for 12 h. After cooling, the precipitates formed were collected by filtration and washed by Soxhlet extraction in ethanol for another 12 h. Afterward, the solid was dried by vacuum evaporation for 24 h to afford the monomer TT–BN (1081 mg) as a light-yellow powder in a 53% isolated yield. $^1H$ NMR (500 MHz, $CF_3COOD$): $\delta$ (ppm) = 8.46 (4H, d, $J$ = 8.2 Hz, Ph $C_2$-$H$), 8.25 (4H, d, $J$ = 8.2 Hz, Ph $C_3$-$H$). $^{13}C$-NMR (125 MHz, $CF_3COOD$): $\delta$ (ppm) = 134.79, 130.12, 127.18, 117.78, 115.52, 113.27, 111.02. MALDI-TOF MS: calcd. $m/z$ = 344.41, found $m/z$ = 344.02.

### Synthesis of 2,5-diphenylthiazolo[5,4-d]thiazole (TT–Bz)

Dithiooxamide (361 mg, 3.0 mmol) and benzaldehyde (0.608 mL, 6.0 mmol) were mixed in DMF (10 mL) in a 100-mL Shrek tube under the nitrogen atmosphere in a glove box. The tube was sealed with a rubber stopper and heated at 150 °C for 24 h. After cooling, the product was collected by filtration, washed with diethyl ether, and recrystallized from dichloromethane to obtain the analog TT–Bz (590 mg) as a pale-yellow powder in a 67% isolated yield. $^1H$ NMR (500 MHz, $CF_3COOD$): $\delta$ (ppm) = 8.22 (4H, d, $J$ = 7.4 Hz, Ph $C_3$-$H$), 7.98 (2H, t, $J$ = 7.5 Hz, Ph $C_1$-$H$), 7.88 (4H, t, $J$ = 7.9 Hz, Ph $C_2$-$H$). $^{13}C$-NMR (125 MHz, $CF_3COOD$): $\delta$ (ppm) = 133.65, 127.67, 117.81, 115.57, 113.31, 111.06. MALDI-TOF MS: calcd. $m/z$ = 294.03, found $m/z$ = 294.39.

### Synthesis of TT–CTP

To a 10-mL Shrek tube containing 100 mg of monomer TT–BN was added dropwise TfOH (1 mL) under the nitrogen atmosphere. The solution was stirred at 263 K for 1.5 h and at 333 K for another 12 h. After cooling to room temperature, a 2-M NaOH aqueous solution was added to the reaction for quenching. The precipitates were washed three times with 10 mL NMP, 2 M NaOH, and deionized water, respectively. The solid was Soxhlet extracted with $CH_2Cl_2$ (12 h), tetrahydrofuran (24 h), and ethanol (12 h). The product was dried at 353 K under vacuum to afford TT–CTP as a yellow powder (91 mg, yield = 91%).

### Synthesis of TTH–CTP

To a solution of 3 mL concentrated HCl, 3 mL $H_2O$ was added to 100 mg of CTP. The suspension was stirred at 30 °C for 4 h. Upon cooling, the mixture was diluted with deionized water (50 mL), and the dark precipitate was collected by filtration, rinsed with water and methanol, and dried under high vacuum to give TTH–CTP.

### Preparation of CTP films

10 mg TTH–CTP and 3 mL formic acid solution was mixed and sonicated for 30 min to obtain a CTP solution with a concentration of 3.3 mg mL$^{-1}$, which was then spin-coated on a $1 \times 1$ cm$^2$ quartz substrate to form a thin film of TTH–CTP with uniform thickness. After annealing at 60°, the TTH–CTP film was immersed in a 20% ammonia water solution for deprotonation and dried in a vacuum oven at 80 °C for 10 h to afford a TT–CTP film with uniform thickness.

## Data availability

The data that support the plots within this paper and other findings of this study are available from the corresponding authors upon reasonable request. Source data are provided in this paper.

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

## Acknowledgements
C.G. acknowledges the financial support by the National Natural Science Foundation of China (grant no. 21975078), the Fundamental Research Funds for the Central Universities, and the start-up foundation of Sichuan University. X.C. acknowledges the financial support by the National Natural Science Foundation of China (grant nos. 21972021 and 22111530111).

## Author contributions
S.W. performed experiments associated with monomer and polymer syntheses and characterizations. Z.X., H.Y., P.Z., and X.C. conducted photocatalytic production of $H_2O_2$ and analyzed the data. D.Z. and H.X. carried out calculation studies. S.F., M.B., and H.I.W. performed THz measurements. Y.W. and Q.L. conducted transient absorption spectroscopy measurements and analyzed the data. C.L. performed electrochemical measurements. C.G. conceived the project and directed the research. All authors contributed to the writing and editing of the manuscript. S.W., Z.X., and D.Z. contributed equally to this work.

## Competing interests
The authors declare no competing interests.
