## [Peer Review File · Nature Communications]

Efficient photocatalytic production of hydrogen peroxide using dispersible and photoactive porous polymersREVIEWER COMMENTS

Reviewer #1 (Remarks to the Author):

This paper describes the preparation of a highly dispersible and photoactive porous photocatalyst based on a covalent triazine framework (CTF) for H₂O₂ production. To solve the dispersion and solution-processing issues of crosslinked porous polymers, the authors proposed the “charge-induced dispersion (CID)” mechanism. Significant photocatalytic H₂O₂ generation can be achieved under ambient conditions at a rate of 23.7 mmol g⁻¹ h⁻¹ with an apparent quantum efficiency of 11.3%. This paper has potential to be published, but the following issue should be considered before publication.

1) Introduction: the more detail background for the development of photocatalyst in H₂O₂ production should be summarized accompanied with relevant literature.

2) Page 6: “TTH-CTF showed an improved performance than TT-CTF due to the superior dispersibility of TTH-CTF in the benzyl alcohol/H₂O system, which facilitated the interfacial homogeneity and boosted the charge transport on the water/TTH-CTF/benzyl alcohol.” Can the authors support this assumption by experimentally. How about the contact angle measurement?

3) Page 7: “The contribution of the catalyst amount to the H₂O₂ concentration revealed volcano-type curves, suggesting the balanced generation and decomposition of H₂O₂.” How much the decomposition rate of H₂O₂?

4) Figure 2b; The authors should check the consumed benzyl alcohol (formed benzaldehyde)/H₂O₂ formation.

5) Page 8: “The total energy of the triazine-protonated structure was lower than that of the TT-protonated structure by 1.032 eV.” On this issue, is it not necessary to consider the solvent effect?

6) Page 9: “These values are quite low compared to most organic semiconductors”. Can the authors comment on the reason?

Reviewer #2 (Remarks to the Author):

This paper reported the efficient photosynthesis of H₂O₂ by designing a highly dispersible and photoactive porous polymer photocatalyst by an in-situ protonation mechanism. The photocatalytic production of H₂O₂ reached an unprecedented generation rate of 23.7 mmol g⁻¹ h⁻¹ and an apparent quantum efficiency of 11.3%, which is far beyond the current best values, thus setting up a new benchmark in this field. It was demonstrated by the spectral characterization that the low exciton binding energy of the porous polymer ensured the effective dissociation of excitons, which, together with the good charge carrier mobility, made the porous polymer a good photocatalyst for H₂O₂ production. The article also explores the specific process of the catalytic reaction through extensive theoretical calculations.

Furthermore, this paper is quite nice in scholarly presentation with solid data. Therefore, considering its novelty, significance, and contribution, I strongly recommend this paper publish in Nature Communications after the following minor revisions:

1. When measuring the gas adsorption of the porous polymers, CO₂ was chosen as the adsorption gas, why not N₂, which is usually used to analyze the pore-size distribution and specific surface area.

2. The experiments of photocatalytic H₂O₂ production were mainly performed on TTH-CTF, but the same experiments (effects of pH and catalyst concentration) on TT-CTF should be also performed to complete the data.
3. Theoretical calculations show that protonation occurs on the triazine ring. What role does the TT moiety play in the solubilization of porous polymers?
4. The supporting information mentions the detection of H₂O₂ concentration by UV-vis spectroscopy, please explain carefully how to detect it and give the standard work curves.
5. Some recent literature regarding triazine-based porous polymers should be cited to benefit the readers, e.g., Chem. Eur. J. 2023, 29, e2022030; Angew. Chem. Int. Ed. 2022, 61, e2021176; Small 2022, 18, 2200984.

Reviewer #3 (Remarks to the Author):

In the present manuscript, Wang and co-workers have reported a covalent triazine polymer for photocatalytic H₂O₂ production. The authors have claimed high photocatalytic H₂O₂ production compared to other organic polymers. To justify the results the authors have done a series of sophisticated experiments. Initially, the manuscript looks interesting to me, however, I am disappointed after reading the result and discussion. The major drawback of the present manuscript is the material synthesis and characterization. Here, I mean proper chemical characterization to have a clear idea about the prepared material. If we don't know what is the exact chemical structure, how many chemical defects are present, and what is the molecular organization, then it is hard to rely on or justify the catalytic performance. This is the case with the present manuscript and hence the present manuscript is not suitable for Nature Communications. In the following section, I have provided my comments and suggestions.

1. First of all, in the title the authors have used "photosynthesis" instead of photocatalysis. I hope the authors are aware of the difference between photosynthesis and photocatalysis. I think for the present manuscript photocatalysis is more appropriate.
2. To me the molecular design looks trivial and I could not find the rationale for designing such a molecule for photocatalytic H₂O₂ production.
3. The authors have leveled the material as a covalent triazine framework, which I do not agree because the analytical results are not sufficient to confirm the partial or complete conversion into the triazine framework. From the dispersity, it seems low degree of polymerization. XPS analysis is required to determine the triazine formation. Also, judging from the absorption and emission, the extent of conjugation is significantly low, and hence the degree of polymerization. Therefore, I am not convinced and skeptical about the chemical structure of the material. Moreover, the resultant material is neither crystalline nor highly porous, and hence it will be a misnomer to call it a framework.
4. In the present manuscript, protonated and non-protonated form of the material displays different optical, electronic and catalytic performance. However, the authors did not comment on the extent of protonation, I mean the material is completely or partially protonated. In principle, there are two possible centers for the protonation triazine and thiazole.

Overall, I doubt the chemical structure and purity of the resultant polymer and hence I have not commented on the optical, electronic, and catalytic performance. Therefore, I request the authors to analyze the polymer critically and optimize the condition for polymer synthesis.

Response to Referee #1

This paper describes the preparation of a highly dispersible and photoactive porous photocatalyst based on a covalent triazine framework (CTF) for H₂O₂ production. To solve the dispersion and solution-processing issues of crosslinked porous polymers, the authors proposed the “charge-induced dispersion (CID)” mechanism. Significant photocatalytic H₂O₂ generation can be achieved under ambient conditions at a rate of 23.7 mmol g⁻¹ h⁻¹ with an apparent quantum efficiency of 11.3%. This paper has potential to be published, but the following issue should be considered before publication.

=====

We appreciate the comments from the reviewer endorsing our work. Regarding the concern raised by the reviewer, we have taken significant consideration of all the comments and suggestions from the reviewer and revised our manuscript accordingly. We believe that the revised manuscript has clarified all the requests and comments from the reviewer.

=====

1) Introduction: the more detail background for the development of photocatalyst in H₂O₂ production should be summarized accompanied with relevant literature.

=====

According to the suggestion from the reviewer, we have added one sentence to the revised manuscript as follows:

“Especially, two-dimensional covalent organic frameworks (COFs)¹⁴⁻²¹ and covalent triazine frameworks (CTFs)²²⁻²⁴ are emerging candidates for efficient photocatalysis because of their predesigned molecular structures and tunable exciton dynamics.”

We have added the following literature to the revised manuscript as Refs. 14–24:

*“14. Chen, D. et al. Covalent organic frameworks containing dual O₂ reduction centers for overall photosynthetic hydrogen peroxide production. Angew. Chem. Int. Ed. **62**, e202217479 (2023).*

*15. Das, P., Chakraborty, G., Roeser, J., Vogl, S., Rabeah, J. & Thomas, A. Integrating bifunctionality and chemical stability in covalent organic frameworks via one-pot multicomponent reactions for solar-driven H₂O₂ production. J. Am. Chem. Soc. **145**, 2975–2984 (2023).*

16. Kou, M. et al. *Molecularly engineered covalent organic frameworks for hydrogen peroxide photosynthesis. Angew. Chem. Int. Ed.* **61**, e202200413 (2022).

17. Zhi, Q. et al. *Piperazine-linked metalphthalocyanine frameworks for highly efficient visible-light-driven H₂O₂ photosynthesis. J. Am. Chem. Soc.* **144**, 21328–21336 (2022).

18. Sun, J. et al. *Pyrene-based covalent organic frameworks for photocatalytic hydrogen peroxide production. Angew. Chem. Int. Ed.* **62**, e202216719 (2023).

19. Das, P., Roeser, J. & Thomas, A. *Solar light driven H₂O₂ production and selective oxidations using a covalent organic framework photocatalyst prepared by a multicomponent reaction. Angew. Chem. Int. Ed. doi: 10.1002/anie.202304349* (2023).

20. Krishnaraj, C. et al. *Strongly reducing (diarylamino)benzene-based covalent organic framework for metal-free visible light photocatalytic H₂O₂ generation. J. Am. Chem. Soc.* **142**, 20107–20116 (2020).

21. Luo, Y. et al. *Sulfone-modified covalent organic frameworks enabling efficient photocatalytic hydrogen peroxide generation via one-step two-electron O₂ reduction. Angew. Chem. Int. Ed.* **62**, e202305355 (2023).

22. Wang, H., Yang, C., Chen, F., Zheng, G. & Han, Q. *A crystalline partially fluorinated triazine covalent organic framework for efficient photosynthesis of hydrogen peroxide. Angew. Chem. Int. Ed.* **61**, e202202328 (2022).

23. Yang, C., Wan, S., Zhu, B., Yu, J. & Cao, S. *Calcination-regulated microstructures of donor-acceptor polymers towards enhanced and stable photocatalytic H₂O₂ production in pure water. Angew. Chem. Int. Ed.* **61**, e202208438 (2022).

24. Cheng, H., Lv, H., Cheng, J., Wang, L., Wu, X. & Xu, H. *Rational design of covalent heptazine frameworks with spatially separated redox centers for high-efficiency photocatalytic hydrogen peroxide production. Adv. Mater.* **34**, 2107480 (2022).”

=====

2) Page 6: “TTH–CTF showed an improved performance than TT–CTF due to the superior dispersibility of TTH–CTF in the benzyl alcohol/H₂O system, which facilitated the interfacial homogeneity and boosted the charge transport on the water/TTH–CTF/benzyl alcohol.” Can the authors support this assumption by experimentally. How about the contact angle measurement?

=====

According to the suggestion from the reviewer, we have measured the contact angles for TT-CTF and TTH-CTF, which were 33.7° and 21.3° , respectively. This result indicated a more hydrophilic nature of TTH-CTF than TT-CTF, which facilitated the interfacial homogeneity and boosted the charge transport on the water/TTH-CTF/benzyl alcohol. We have added this result to the revised Supplementary Materials as Figure S34.

Supplementary Figure S34. Contact-angle measurements for (a) TT-CTF and (b) TTH-CTF thin films.

=====

3) Page 7: “The contribution of the catalyst amount to the H_2O_2 concentration revealed volcano-type curves, suggesting the balanced generation and decomposition of H_2O_2 .” How much the decomposition rate of H_2O_2 ?

=====

We thank the reviewer for this constructive suggestion. Accordingly, we measured the H_2O_2 decomposition rates using TT-CTF and TTH-CTF as the photocatalysts, respectively. We have added the results to the revised Supplementary Materials as Figure S38.

Supplementary Figure S38. H₂O₂ decomposition rates along with reaction time. Taking into account the acceleration of H₂O₂ decomposition under irradiation, the stability of the produced H₂O₂ was assessed by measuring the degradation behavior of H₂O₂ generated during the reaction of the prepared samples. The mixed solution of H₂O₂ (1 mM, 20 mL) and photocatalyst (1 mg mL⁻¹) was sonicated for 2 minutes, followed by purging the system with argon. Subsequently, the light source was turned on, and the residual H₂O₂ concentration was measured every 15 minutes to assess its stability.

=====

4) Figure 2b; The authors should check the consumed benzyl alcohol (formed benzaldehyde)/H₂O₂ formation.

=====

We thank the reviewer for this constructive suggestion. Accordingly, we measured the concentration of produced benzaldehyde using gas chromatography, which revealed that the produced H₂O₂ and benzaldehyde were almost equimolar, demonstrating that the half reaction of H₂O₂ generation was the oxidation of benzyl alcohol to benzaldehyde. We have added this result to the revised Supplementary Materials as Figure S33.

Supplementary Figure S33. (a) Gas chromatography data of the produced benzaldehyde. (b) Concentrations of H₂O₂ and benzaldehyde in the photocatalytic full reaction of TT-CTF and TTH-CTF.

5) Page 8: “The total energy of the triazine-protonated structure was lower than that of the TT-protonated structure by 1.032 eV.” On this issue, is it not necessary to consider the solvent effect?

We appreciate the reviewer’s comments, which helped us improve the quality of our work. The catalytic reactions of TT-CTF were carried out in the aqueous system, so all theoretical calculations in this work have considered the solvation effect and performed using SMD implicit solvation model in Gaussian 16. We included the adoption of the SMD solvation model in the SI but did not emphasize it in the main manuscript, which may cause misunderstanding to the reviewer.

6) Page 9: “These values are quite low compared to most organic semiconductors”. Can the authors comment on the reason?

We thank the reviewer for this thought-provoking comment. Indeed, the design of low- E_b organic semiconducting materials is still a challenge, which is because many intricate factors such as molecular structures and aggregation states influence the E_b value of the materials, rendering the prediction of E_b of a material before synthesis rather difficult. Nevertheless, a

few consensuses have been reached, such as the integration of strong donor-acceptor moieties into one molecule, which has been proven to be effective in polymer photovoltaic materials and some covalent organic frameworks (COFs). However, the TT and triazine (and its protonated counterpart) moieties in our materials are all n-type (thus acceptors), which is exceptional to yield such low- E_b materials. Currently, we are not able to provide a full explanation of these low- E_b CTFs, although the data have been repeated several times to confirm their validity. However, with the reviewer's comment in mind, we have started to collaborate with theoretical chemists to seek the underlying mechanism, and we would like to report the design principles of low- E_b porous polymers in future manuscripts.

We sincerely hope that the reviewer could understand our favorable attitude regarding the revision and be satisfied with the revision we made. We appreciate the reviewer for the above constructive comments that improve the manuscript.

=====

Response to Referee #2

This paper reported the efficient photosynthesis of H₂O₂ by designing a highly dispersible and photoactive porous polymer photocatalyst by an in-situ protonation mechanism. The photocatalytic production of H₂O₂ reached an unprecedented generation rate of 23.7 mmol g⁻¹ h⁻¹ and an apparent quantum efficiency of 11.3%, which is far beyond the current best values, thus setting up a new benchmark in this field. It was demonstrated by the spectral characterization that the low exciton binding energy of the porous polymer ensured the effective dissociation of excitons, which, together with the good charge carrier mobility, made the porous polymer a good photocatalyst for H₂O₂ production. The article also explores the specific process of the catalytic reaction through extensive theoretical calculations. Furthermore, this paper is quite nice in scholarly presentation with solid data. Therefore, considering its novelty, significance, and contribution, I strongly recommend this paper publish in Nature Communications after the following minor revisions:

=====

We appreciate the comments from the reviewer endorsing our work. Regarding the concern raised by the reviewer, we have taken significant consideration of all the comments and suggestions from the reviewer and revised our manuscript accordingly. We believe that the revised manuscript has clarified all the requests and comments from the reviewer.

=====

1. When measuring the gas adsorption of the porous polymers, CO₂ was chosen as the adsorption gas, why not N₂, which is usually used to analyze the pore-size distribution and specific surface area.

=====

We appreciate the reviewer for this comment. N₂-sorption is a standard method to characterize the porosity of a material. However, in the following two cases, the N₂-sorption is not adaptable for characterizing the porosity: 1) ultramicroporous materials whose pore apertures are smaller than the kinetic diameter of N₂; 2) microporous materials with highly polar pore environments that result in low affinity with non-polar N₂. In these two cases, CO₂ is usually selected as the adsorbate to detect the porosity. Our case is both small pore size and ionic species that render **TTH-CTF** not adsorbing N₂, therefore, we employed CO₂-sorption measurements to evaluate the porosity of our materials.

2. The experiments of photocatalytic H_2O_2 production were mainly performed on TTH-CTF, but the same experiments (effects of pH and catalyst concentration) on TT-CTF should be also performed to complete the data.

According to the suggestion from the reviewer, we have measured the same experiments using TT-CTF as the photocatalyst, which exhibited the same trend as the TTH-CTF photocatalyst. We have added the results to the revised Figure 2e and Figure S37.

Figure 2 | Photocatalytic production of H_2O_2 . **a**, Time-dependent production rate of H_2O_2 on TT-CTF and TTH-CTF in pure water. **b**, H_2O_2 generation rates of photocatalytic half reaction of TT-CTF and TTH-CTF with and without benzyl alcohol as a hole sacrificial agent. **c**, UV-vis spectra and AQY of H_2O_2 generation for TT-CTF and TTH-CTF. **d**, H_2O_2 generation rates and AQY of different photocatalysts. The red star represents TTH-CTF in this work (with benzyl alcohol, see panel b) and the orange, green, sky-blue, and blue circles correspond to other previously reported photocatalysts. **e**, H_2O_2 generation rates of photocatalytic half-reaction of TT-CTF and TTH-CTF under different pH values. **f**, H_2O_2 generation rates

of photocatalytic half-reaction of **TTH-CTF** under different cycles.

Supplementary Figure S37. H₂O₂ generation rates of photocatalytic half reaction of **TT-CTF** and **TTH-CTF** under different photocatalyst concentrations.

3. Theoretical calculations show that protonation occurs on the triazine ring. What role does the TT moiety play in the solubilization of porous polymers?

We appreciate the reviewer for this comment. Currently, there have been some reports on the dispersible CTFs (e.g., *CCS Chem.* **2020**, 2, 2688; *Chem. Mater.* **2021**, 33, 3386; *Mater. Horiz.* **2021**, 8, 3088), in which several electron-donating or electron-withdrawing groups can promote the protonation of the triazine rings. However, the protonated CTFs with strong electron-donating groups are difficult to completely deprotonate, rendering the residual protons always on the polymer skeletons that inevitably influence their performances. By contrast, in this manuscript, we employed a weak n-type TT moiety to increase the polarity of the CTF and eventually facilitate the protonation of the triazine rings. More importantly, the weak electron-withdrawing TT moiety makes the protonation/deprotonation of the triazine rings highly reversible, thus allowing us to individually control the structures and properties of the neutral and protonated CTFs.

4. The supporting information mentions the detection of H₂O₂ concentration by UV-vis spectroscopy, please explain carefully how to detect it and give the standard work curves.

According to the suggestion from the reviewer, we have added the working curve and the detailed experimental information to the revised Supplementary Materials as Fig. S29.

Supplementary Figure S29. Working curve of the titanium sulfate spectrophotometry. Based on the stable orange-yellow complex (410 nm maximum absorption wavelength) formed by H₂O₂ with Ti⁴⁺ ion, the content of H₂O₂ was analyzed by titanium sulfate spectrophotometry. After the reaction, the reaction liquid is collected by filtration. The reaction liquid (5 mL) to be measured was added to 2 mL of the prepared titanium sulfate Ti(SO₄)₂ solution and then transferred to a 25 mL volumetric bottle for constant volume. The absorbance of part of the solution was measured by UV spectrophotometer.

5. Some recent literature regarding triazine-based porous polymers should be cited to benefit the readers, e.g., Chem. Eur. J. 2023, 29, e202203077; Angew. Chem. Int. Ed. 2022, 61, e202117668; Small 2022, 18, 2200984.

We thank the reviewer for this constructive suggestion. According to the suggestion from the reviewer, we have cited the above pioneer works as Refs. 11–13 in the revised manuscript.

We sincerely hope that the reviewer could understand our favorable attitude regarding the revision and be satisfied with the revision we made. We appreciate the reviewer for the above constructive comments that improve the manuscript.

=====

Response to Referee #3

In the present manuscript, Wang and co-workers have reported a covalent triazine polymer for photocatalytic H₂O₂ production. The authors have claimed high photocatalytic H₂O₂ production compared to other organic polymers. To justify the results the authors have done a series of sophisticated experiments. Initially, the manuscript looks interesting to me, however, I am disappointed after reading the result and discussion. The major drawback of the present manuscript is the material synthesis and characterization. Here, I mean proper chemical characterization to have a clear idea about the prepared material. If we don't know what is the exact chemical structure, how many chemical defects are present, and what is the molecular organization, then it is hard to rely on or justify the catalytic performance. This is the case with the present manuscript and hence the present manuscript is not suitable for Nature Communications. In the following section, I have provided my comments and suggestions.

=====

We appreciate the comments from the reviewer endorsing the interests of our work. Regarding the concern raised by the reviewer, we have taken significant consideration of all the comments and suggestions from the reviewer and revised our manuscript accordingly. We believe that the revised manuscript has clarified all the requests and comments from the reviewer.

=====

1. First of all, in the title the authors have used "photosynthesis" instead of photocatalysis. I hope the authors are aware of the difference between photosynthesis and photocatalysis. I think for the present manuscript photocatalysis is more appropriate.

=====

We are sorry for making confusion. According to the suggestion from the reviewer, we have changed the title to "Efficient photocatalytic production of hydrogen peroxide using dispersible and photoactive porous polymers" in the revised manuscript.

=====

2. To me the molecular design looks trivial and I could not find the rationale for designing such a molecule for photocatalytic H₂O₂ production.

=====

We appreciate this thought-provoking comment from the reviewer. However, please let us explain why our materials are special for photocatalysis. As mentioned in the manuscript, one critical issue in photocatalytic H₂O₂ production and other water-splitting approaches is the mismatched interfaces between water and catalysts. When using insoluble and unprocessable POP materials, two critical issues are presented, namely, (1) the low water-catalyst affinity resulting in imperfect contact and thus interfacial resistance that impedes the electron transport, and (2) small amounts of catalytic centers that expose to water due to the bulk aggregation of the POP powders. Therefore, even using crystalline POP materials such as COFs for photocatalytic water splitting, a general protocol is to exfoliate the bulky COF powders to nanosheets by sonication, to expose more catalytic centers. However, this does not fundamentally solve the issue of interfacial resistance. Therefore, developing solution-processed POP materials is essential not only for improving the catalytic activity but also for many mass-transportive applications. This is why our group insists to develop solution-processed POPs in the past five years (*Acc. Mater. Res.* **2022**, *3*, 1049). Back to this manuscript, the uniqueness of this material lies in the following three points. (1) We encoded the TT moiety, which is a weak electron acceptor with large polarity, to the CTF material to promote the protonation of the triazine rings, thus endowing the CTF material with solution dispersibility for minimizing the interfacial resistance between water and catalysts. (2) In the previous reports from our group and other groups (e.g., *CCS Chem.* **2020**, *2*, 2688; *Chem. Mater.* **2021**, *33*, 3386; *Mater. Horiz.* **2021**, *8*, 3088), they used electron donor moieties to promote protonation. However, these protonated CTFs are difficult to completely deprotonate, rendering the residual protons always on the polymer skeletons that inevitably influence their performances. By contrast, the weak electron acceptor TT moiety makes the protonation/deprotonation of the triazine rings highly reversible, thus allowing us to individually control the structures and properties of the neutral and protonated CTFs. (3) The CTFs possess exceptional photoactivity in terms of low E_b values, ultrafast exciton/charge-carrier generation, and high charge-carrier mobilities, which ensure efficient photocatalytic H₂O₂ production from both thermodynamic and kinetic aspects. Therefore, our material, although looks “simple”, possesses both solution processability and photoactivity, which synergistically boost the H₂O₂ synthesis and achieve unprecedented catalytic performance. Therefore, the key design rationale is to develop solution-dispersible porous polymers.

=====

3. The authors have leveled the material as a covalent triazine framework, which I do not agree because the analytical results are not sufficient to confirm the partial or complete conversion into the triazine framework. From the dispersity, it seems low degree of polymerization. XPS analysis is required to determine the triazine formation. Also, judging from the absorption and emission, the extent of conjugation is significantly low, and hence the degree of polymerization. Therefore, I am not convinced and skeptical about the chemical structure of the material. Moreover, the resultant material is neither crystalline nor highly porous, and hence it will be a misnomer to call it a framework.

=====

Please let us explain why we believe that we have successfully synthesized the CTFs. The reviewer mentioned that “from the dispersity, it seems low degree of polymerization.” This is a traditional idea that polymers with low polymerization degrees exhibit good dispersibility. However, our systems are conceptionally different from traditional polymers. We incorporated plenty of charges onto the skeletons of the porous polymers to enhance their interactions with solvents, leading to significant dispersibility for solution processing. Indeed, this concept was first suggested by us in 2019 (*Chem. Sci.* **2019**, *10*, 1023), and then we developed several solution-processed systems including crystalline COFs and amorphous POPs (*J. Am. Chem. Soc.* **2022**, *144*, 8961; *CCS Chem.* **2020**, *2*, 2688; *Mater. Horiz.* **2021**, *8*, 3088; *Acc. Mater. Res.* **2022**, *3*, 1049). In all these systems, the COF nanosheets or POP nanoparticles have large sizes, namely high molecular weight and polymerization degree. We call this strategy the “charge-induced dispersion (CID)” strategy. Therefore, we believe that the high polymerization degree and high dispersibility of the CTFs in this work are not in conflict. Of course, according to the suggestion from the reviewer, we have performed the following additional experiments to clarify the successful formation of CTFs:

(1) We measured the XPS spectra of **TT-CTF**. In the N1s spectra of **TT-CTF**, the peak 2 attributing to the unreacted cyano group was only 6.2% of the total N atoms, demonstrating the successful formation of the triazine rings and the high polymerization degree. We have added the XPS results to the revised Supplementary Materials as Fig. S17 and Table S3.

Supplementary Figure S17. XPS analysis of **TT-CTF** and **TTH-CTF**. (a) **TT-CTF**, C1s; (b) **TT-CTF**, N1s; (c) **TT-CTF**, S2p; (d) **TTH-CTF**, C1s; (e) **TTH-CTF**, N1s; (f) **TTH-CTF**, S2p.

In the N1s spectra of **TT-CTF**, the peak 2 attributing to the unreacted cyano group was only 6.2% of the total N atoms, demonstrating the successful formation of the triazine rings and the high polymerization degree. The N peaks attributing to the thiazole and triazine merged into one peak at 396 eV. On the other hand, in **TTH-CTF**, peaks 1 and 2 were attributed to the thiazole and triazine groups, respectively, whereas peak 3 was attributed to the protonated triazine group. The ratio of peaks 1, 2, and 3 was 1.4:2.2:1, which means that the protonated N atom was 31.3% of the total N atoms in one triazine ring. Therefore, the protonation degree was 93.8% (one triazine ring contains nearly one positive charge). Additionally, the S2p spectra remained almost unchanged before and after protonation, which further indicated that the protonation occurred on the triazine rather than thiazole.

Supplementary Table S3. XPS analysis results of **TT-CTF** and **TTH-CTF**.

Sample	Element	Peak	Concentration (%)	
TT-CTF	C	Peak 1	43.7	76.4 (in total)
		Peak 2	22.0	
		Peak 3	10.7	
	N	Peak 1	15.1	16.1 (in total)
		Peak 2	1.0	
	S	Peak 1	4.8	7.5 (in total)
Peak 2		2.7		
TTH-CTF	C	Peak 1	38.4	76.9 (in total)
		Peak 2	28.8	
		Peak 3	9.7	
	N	Peak 1	4.8	15.4 (in total)
		Peak 2	7.3	
		Peak 3	3.3	
S	Peak 1	4.6	7.7 (in total)	
	Peak 2	3.1		

(2) We measured the absorption and emission spectra of the monomer, TT-BN, which exhibited the absorption onset and the maximum emission peaks at 478 and 498 nm, respectively. Compared to those of the CTFs, the monomer showed substantially blue-shifted absorption and emission and thus indicated low conjugation of the monomer compared to CTFs. We have added the absorption and emission spectra of the monomer to Figs. S20 and S23 of the revised Supplementary Materials.

Supplementary Figure S20. (a) UV-vis spectra of TT-BN, TT-CTF, and TTH-CTF. (b) Tauc plot of TT-CTF and TTH-CTF.

Supplementary Figure S23. Photoluminescence spectra of TT-BN, TT-CTF, and TTH-CTF in the air.

(3) Regarding the term “covalent triazine framework”, we fully agree with the reviewer that an amorphous triazine polymer may be not suitable to call a framework. However, please let us explain why we finally used this term. Since the first CTF (CTF-1) was developed, several methods have been developed to synthesize CTFs, including ionothermal synthesis, triflic acid-catalyzed synthesis, Friedel–Crafts reaction, and polycondensation of aldehyde and imidamide monomers. Based on these methods, only very few CTFs were reported to show crystallinity, whereas most CTFs were amorphous. However, because CTF-1 was crystalline and was termed “framework”, the following CTF materials followed this term even though they were not crystalline and/or highly porous. We are clear that it’s better to term “covalent triazine polymer” rather than “covalent triazine framework”, but herein we would like to follow the conventional “CTF” term because we believe that the novelty of our work is not to create a new triazine material but the solution-processing concept which is also potential to extend to other CTF materials (e.g., *Mater. Horiz.* **2021**, 8, 3088). Therefore, we sincerely hope that the reviewer could understand the novelty of our work and agree with the term we used.

4. In the present manuscript, protonated and non-protonated form of the material displays different optical, electronic and catalytic performance. However, the authors did not comment on the extent of protonation, I mean the material is completely or partially protonated. In principle, there are two possible centers for the protonation triazine and thiazole.

According to the suggestion from the reviewer, we have measured the XPS spectra of **TT-CTF** and **TTH-CTF**. In the N1s spectra of **TT-CTF**, the N peaks attributing to the thiazole and triazine merged into one peak at 396 eV. On the other hand, in **TTH-CTF**, peaks 1 and 2 were attributed to the thiazole and triazine groups, respectively, whereas peak 3 was attributed to the protonated triazine group. The ratio of peaks 1, 2, and 3 was 1.4:2.2:1, which means that the protonated N atom was 31.3% of the total N atoms in one triazine ring. Therefore, the protonation degree was 93.8% (one triazine ring contains nearly one positive charge). Additionally, the S2p spectra remained almost unchanged before and after protonation, which further indicated that the protonation occurred on the triazine rather than thiazole. We have added the XPS results to the revised Supplementary Materials as Fig. S17 and Table S3. We have also added the following sentence to the revised manuscript:

“The protonation degree in *TTH-CTF* was 93.8% according to the X-ray photoelectron spectroscopy (XPS) analysis (Supplementary Fig. S17, Table S3).”

Supplementary Figure S17. XPS analysis of *TT-CTF* and *TTH-CTF*. (a) *TT-CTF*, C1s; (b) *TT-CTF*, N1s; (c) *TT-CTF*, S2p; (d) *TTH-CTF*, C1s; (e) *TTH-CTF*, N1s; (f) *TTH-CTF*, S2p.

In the N1s spectra of *TT-CTF*, the peak 2 attributing to the unreacted cyano group was only 6.2% of the total N atoms, demonstrating the successful formation of the triazine rings and the high polymerization degree. The N peaks attributing to the thiazole and triazine merged into one peak at 396 eV. On the other hand, in *TTH-CTF*, peaks 1 and 2 were attributed to the thiazole and triazine groups, respectively, whereas peak 3 was attributed to the protonated triazine group. The ratio of peaks 1, 2, and 3 was 1.4:2.2:1, which means that the protonated N atom was 31.3% of the total N atoms in one triazine ring. Therefore, the protonation degree was 93.8% (one triazine ring contains nearly one positive charge). Additionally, the S2p spectra remained almost unchanged before and after protonation, which further indicated that the protonation occurred on the triazine rather than thiazole.

Supplementary Table S3. XPS analysis results of *TT-CTF* and *TTH-CTF*.

Sample	Element	Peak	Concentration (%)	
TT-CTF	C	Peak 1	43.7	76.4 (in total)
		Peak 2	22.0	
		Peak 3	10.7	
	N	Peak 1	15.1	16.1 (in total)
		Peak 2	1.0	
	S	Peak 1	4.8	7.5 (in total)
Peak 2		2.7		
TTH-CTF	C	Peak 1	38.4	76.9 (in total)
		Peak 2	28.8	
		Peak 3	9.7	
	N	Peak 1	4.8	15.4 (in total)
		Peak 2	7.3	

S	Peak 3	3.3	7.7 (in total)
	Peak 1	4.6	
	Peak 2	3.1	

=====
Overall, I doubt the chemical structure and purity of the resultant polymer and hence I have not commented on the optical, electronic, and catalytic performance. Therefore, I request the authors to analyze the polymer critically and optimize the condition for polymer synthesis.
=====

According to the suggestion from the reviewer, we have performed additional experiments and revised our manuscript critically. We believe the revision has clarified all the issues raised by the reviewer especially the structure and purity of the polymer. We sincerely hope the reviewer could understand our positive situation regarding the revision and be satisfied with the additional experiments and revisions we performed. We appreciate the reviewer for the above constructive comments that improve the manuscript.
=====

REVIEWER COMMENTS

Reviewer #1 (Remarks to the Author):

The authors have carried out further studies on the system and clarified the original questions. The manuscript reads well and now this paper is recommended to publish as revised version.

Reviewer #2 (Remarks to the Author):

The authors have addressed all my concerns, and the paper can be accepted with present form.

Reviewer #3 (Remarks to the Author):

I would like to thank the authors for revising the manuscript and making it more scientifically convincing. I carefully read their response, however still skeptical about the chemical structure of the material. From the XPS results, it seems like the polymer has a random chemical structure and contains lots of defects. The XPS spectra of N1s is fitted with two- or three- peaks having different full width at half maxima (FWHM) which is not recommended and not reliable. Peak 1 and peak 2 of TTH-CTF appear at the same binding energy whereas they show different binding energy for the TT-CTF. Also, the protonated triazine peak 3 is below the noise level and is hard to consider. In general, the protonation of N, significantly changes the binding energy, and a distinct peak can be observed in XPS. The C1s spectrum of the TTH-CTF is quite broad compared to the TT-CTF. Also, in general, it's difficult to quantify the chemical composition from XPS due to the chance of dust contamination. Hence, I think the XPS results are inconclusive. Authors can perform CHN analysis to determine the chemical composition. Instead of absorption and emission, which might be indirect proof for the degree of polymerization, authors are requested to perform some experiments to comment on the degree of polymerization.

Regarding the name, still I prefer to call it as "covalent triazine polymer", instead of covalent triazine framework when there is no crystallinity and high surface area, although several research groups misnamed it. It's always better to use right nomenclature for the chemicals. Overall, if I consider it as "covalent triazine polymer" having chemical defects, then I am ok with all the analysis and catalytic results. However, in that case, the structure-property relationship cannot be established or generalized.

Response to Referee #1

The authors have carried out further studies on the system and clarified the original questions. The manuscript reads well and now this paper is recommended to publish as revised version.

=====

We appreciate this comment.

=====

Response to Referee #2

The authors have addressed all my concerns, and the paper can be accepted with present form.

=====

We appreciate this comment.

=====

Response to Referee #3

I would like to thank the authors for revising the manuscript and making it more scientifically convincing. I carefully read their response, however still skeptical about the chemical structure of the material. From the XPS results, it seems like the polymer has a random chemical structure and contains lots of defects. The XPS spectra of N1s is fitted with two- or three- peaks having different full width at half maxima (FWHM) which is not recommended and not reliable. Peak 1 and peak 2 of TTH-CTF appear at the same binding energy whereas they show different binding energy for the TT-CTF. Also, the protonated triazine peak 3 is below the noise level and is hard to consider. In general, the protonation of N, significantly changes the binding energy, and a distinct peak can be observed in XPS. The C1s spectrum of the TTH-CTF is quite broad compared to the TT-CTF. Also, in general, it's difficult to quantify the chemical composition from XPS due to the chance of dust contamination. Hence, I think the XPS results are inconclusive. Authors can perform CHN analysis to determine the chemical composition. Instead of absorption and emission, which might be indirect proof for the degree of polymerization, authors are requested to perform some experiments to comment on the degree of polymerization.

=====

We appreciate the reviewer for endorsing the revision of our manuscript, and we also thank the reviewer for his/her constructive comments that improve the quality of the manuscript.

We fully agree with the reviewer that the XPS results were inconclusive because of the potential contamination of the samples. Therefore, to make the manuscript rigorous, we deleted the XPS results (including the figure and table) in the revised Supplementary Materials and their discussion in the revised manuscript.

Indeed, we have involved the elemental analysis in the initial submitted manuscript (Supplementary Table S2). However, the elemental analysis was performed by the technicians in our institute, and they never performed the measurements immediately after the activation of our samples. As the climate of our city is perennially humid, the samples easily adsorb moisture and make the elemental analysis a bit inaccurate. Therefore, we tend to not use the elemental analysis results to evaluate the polymerization degree. Instead, we use the IR spectra to comment on the polymerization degree. As shown in Supplementary Figure S10, the peaks at 1600 cm^{-1} in both TT-BN monomer and TT-CTP were attributed to the C=N vibration peak

in the thiazolo[5,4-*d*]thiazole (TT) moiety, whereas the bands at 2238 cm⁻¹ in both TT–BN monomer and TT–CTP were assigned to the vibration of cyano groups. The cyano vibrational peak in TT–CTP almost disappeared, so if we normalize the intensity of the 1600 cm⁻¹ peak and check the ratio of the 2238 cm⁻¹ peak in the TT–BN monomer and TT–CTP, we could roughly estimate the polymerization degree to be over 98%. Of course, we know that it is difficult to precisely evaluate the polymerization degree at the current stage because of the amorphous nature of our materials. However, with the reviewer’s comments in mind, we have endeavored to start the synthesis of the crystalline materials even though such synthesis is rather tough for us. We would like to report the crystalline CTFs with their precise structures and polymerization degrees for photocatalysis in future manuscripts. We sincerely hope that the reviewer can understand the difficulty in characterizing the polymerization degree of amorphous crosslinked polymers and is satisfied with the revision we made.

According to the suggestion from the reviewer, we have added one sentence in the revised manuscript to comment on the polymerization degree:

“The polymerization degree of TT–CTP was estimated to be over 98% (Supplementary Fig. S10).”

=====

Regarding the name, still I prefer to call it as “covalent triazine polymer”, instead of covalent triazine framework when there is no crystallinity and high surface area, although several research groups misnamed it. It’s always better to use right nomenclature for the chemicals.

=====

We thank the reviewer for this constructive comment. According to the suggestion from the reviewer, we have revised all the “CTF” with “CTP” throughout the revised manuscript.

=====

Overall, if I consider it as “covalent triazine polymer” having chemical defects, then I am ok with all the analysis and catalytic results. However, in that case, the structure-property relationship cannot be established or generalized.

=====

We thank the reviewer for endorsing all the analysis and catalytic results in our manuscript. The highlight and the novelty of this manuscript is the “soluble” CTP that minimizes the interfacial mismatch to improve the photocatalytic performance. This manuscript is just the beginning of this direction in our group, and we are focusing more solution-processable porous materials for catalytic applications. With the reviewer’s comments in mind, we will try our best to develop more powerful systems and gain in-depth insight into the structure-property relationship.

We sincerely hope the reviewer can understand our positive situation regarding the revision and be satisfied with the revisions we performed. We appreciate the reviewer for the above constructive comments that improve the manuscript.

=====

REVIEWERS' COMMENTS

Reviewer #3 (Remarks to the Author):

In the revised manuscript, the authors have tried to address most of the comments. Final two comments before accepting this manuscript

- 1) Please include the XPS results in the supporting information with the correct fitting.
- 2) In general, FT-IR is a qualitative method and hence the authors are requested to remove the claim of 98% polymerization.

Response to Referee #3

In the revised manuscript, the authors have tried to address most of the comments. Final two comments before accepting this manuscript

1) Please include the XPS results in the supporting information with the correct fitting.

2) In general, FT-IR is a qualitative method and hence the authors are requested to remove the claim of 98% polymerization.

=====

According to the suggestion from the reviewer, we have added the XPS results to the revised Supplementary Materials as Fig. S17 and Table S3. We also deleted the claim of 98% polymerization from the revised manuscript.

We sincerely hope the reviewer can understand our positive situation regarding the revision and be satisfied with the revisions we performed. We appreciate the reviewer for the above constructive comments that improve the manuscript.

=====